ecology/microbiology

*Coptotermes curvignathus*, by oral-contact, by contact, bacterial concentration, biocontrol efficiency

**Authors for correspondence:**
Kit Ling Chin
e-mail: kitling.chin419@gmail.com
Paik San H'ng
e-mail: ngpaiksan@gmail.com

†Institute of Tropical Forestry and Forest Products, Universiti Putra Malaysia, 43400 UPM Serdang, Selangor, Malaysia.

# Septicaemia of subterranean termites *Coptotermes curvignathus* caused by disturbance of bacteria isolated from termite gut and its foraging pathways

Kit Ling Chin[1], Paik San H'ng[1,2,†], Wan Zhen Wong[1], Chuan Li Lee[1], Pui San Khoo[1], Abdullah Chuah Luqman[1], Zaidon Ashaari[2] and Seca Gandaseca[2]

[1]Institute of Tropical Forestry and Forest Product, and [2]Faculty of Forestry and Environment, Universiti Putra Malaysia, 43400 Serdang, Selangor, Malaysia

CLL, 0000-0002-2293-4088; PSK, 0000-0002-8584-9198

Microbial pathogens continue to attract a great deal of attention to manage the termite population. Every bacterium has its own mode of action and in fact, the mechanisms used by bacteria to attack termites remain elusive at the moment. Hence, the objective of this study was to evaluate the susceptibility of subterranean termites *Coptotermes curvignathus* to opportunistic pathogens using culturable aerobic bacteria isolated from the termite gut and its foraging pathways. Bacterial suspensions were prepared in concentrations of $10^3$, $10^6$ and $10^9$ colony-forming units (CFU) ml$^{-1}$ and introduced to the termites via oral-contact and physical contact treatment. The data show that contact method acted slower and gave lower mortality, compared to the oral-contact method. *Coptotermes curvignathus* were highly susceptible to *Serratia marcescens* and *Pseudomonas aeruginosa*. *Serratia marcescens* showed the highest mortality percentage of 68% and 54% at bacterial concentration of $10^9$ CFU ml$^{-1}$ via oral-contact and contact method, respectively. *Serratia marcescens* was also defined as the bacteria with the highest ability to induce the high mortality of *C. curvignathus* with the lowest concentration of bacterial suspension at a given time under laboratory condition. The results of this study indicate that *P. aeruginosa* and *S. marcescens* in particular may be attractive candidates worth further examination as a possible

biocontrol agent against *C. curvignathus* in the field and to evaluate environmental and ecological risks of the biocontrol.

# 1. Introduction

In recent decades, there has been a steady increase in the amount of pesticides used. The excessive use of pesticides in a variety of pest management situations including commercial farming, lawn care/landscaping, forestry and wood preservation may lead to soil and water pollution and the destruction of biodiversity which are the consequences of pest's habitat destruction by human activity. According to the report from Environmental Protection Agency (EPA) about 5 billion pounds of pesticide's active ingredients were used worldwide and out of that amount, approximately 1 billion pounds are used for termite control. Especially in the warmer regions of the world, termites are the major wood-destroying structural pests and the huge amount of repair costs were spent to suppress wood-feeding termite [1]. In terms of economic loss, around USD 40 billion has been spent annually on repairing the destroyed building of wood, forest and other commercial products responsible by termites [2]. Synthetic pesticide such as imidacloprid, bifenthrin, chlorpyrifos, endosulfan and lindane remains as the primary methods used to prevent and suppress termite attack on wooden materials and plantation [3]. However, the excessive use of chemicals on controlling termites is a serious environmental concern, with many countries under Food and Agriculture Organization (FAO) and United Nations Environment Programme (UNEP) banning utilization of many organochlorines due to harmful to human health [4].

Pursuant to these chemical restrictions and environmental regulation, the focus in termite management research has shifted to biological treatment methods. Biological control (biocontrol) can be defined as the control or reduction of termite damage by reducing the quantity and detrimental activity of termites using the entomopathogenic microorganism or the product resulting from a natural biological process. Biocontrol is an environmentally friendly alternative to the widespread use of chemicals. With successful implementation, it can create pest populations management which is permanent, cost-efficient and with minimal ecological disruption [5].

Microbial pathogens continue to attract a great deal of attention to manage the termite population. The earliest biocontrol used on termites, reported by Beal and Kais in 1962 [6], identified *Aspergillus flavus* link as a fungal pathogen of *Reticulitermes* sp. [7]. However, termite's natural behaviour such as allogrooming, proctodeal and stomodeal trophallaxis, coprophagy and cannibalism of injured nest-mates are protecting them from the harmful pathogen. Typically, infected cadavers are quarantined by covering them with soil or constructing walls around them with fecal pellets [8]. Thus, in most cases of successful biological control on termites, bacteria have to be ingested by termites or horizontally transmitted through coprophagy to cause septicaemia [9]. Ingested bacteria are then transferred to gut through food by mouth and disturb the microbes' ecology of the gut.

Moreover, the interest in bacteria associated with a growing number of newly isolated entomopathogenic nematodes (EPNs), including *Photorhabdus*, *Xenorhabdus* and, more recently, *Serratia*, *Alcaligenes* and other species, is rising [10–12]. Such bacteria, normally released by nematodes inside the body of host insects, produce a variety of toxins and virulence factors impairing insect immune response, and a range of antimicrobial compounds antagonizing resident bacteria [13–15]. Another source of entomopathogenic bacteria is represented by species living in the body of insects that under favourable conditions behave as insect pathogens [16]. Regardless of its origin, any culturable entomopathogenic bacterium would potentially be used as a stand-alone bioinsecticide. This research trend is in line with the present legislative framework of main world regions (i.e. USA and Europe) that fosters the development and use of low-risk biocidal and plant protection products [17]. On the other hand, specific genetic traits of bacterial pathogens could be exploited as plant-incorporated protectants [18]. The purpose of this study is to contribute to knowledge advancements on the oral insecticidal potential of a variety of culturable bacterial species, other than those already in use for pest management. The pathogenicity of new bacterial strains that were isolated during a large screening programme against termite importance is comparatively reported.

Bacteria can sometimes access the haemocoel directly, either through an accidental breaching of the cuticle, or by assisted transport via an entomophagous host, such as a nematode [9]. Other effective biological control is through a contact mechanism using the chitinolytic bacteria. In fact, the mechanisms used by bacteria to attack termites still remain unknown. Every bacterium has its own mode of action. The ability of bacteria to become a death causative agent on termite has not been characterized. Therefore, it is important to find out whether the bacteria kill the target host by ingestion or by contact.

This is because when a host is killed by a pathogen via ingestion, the pathogen should be applied to the host's food. However, if the host can be simply killed by the pathogen via contact, the pathogen can be just applied to the surroundings of the host. As mentioned above, bacteria have a high potential for development as a biological control agent for termites. However, there is a lack of experimental data to support further research. Most of the findings have led to overestimates of the real potential for termite biological control [6]. One of the causes of this result bias is the lack of a standard protocol to determine pathogenicity towards termites. We hypothesize that termites die due to bacteria colonization when any bacterium is adjusted to a sufficiently high concentration. Hence, it was difficult to define as termite entomopathogen, since bacteria become pathogenic when they are in the high-concentration condition. Therefrom, we developed a 'biocontrol score' which can define the level of the pathogenicity of bacteria towards termite. The biocontrol efficiency was defined as the ability of the bacteria to induce high mortality of *C. curvignathus* with the lowest concentration of bacterial suspension at a given time under laboratory condition. Consequently, the aim of the present study was to evaluate the susceptibility of the subterranean termite *C. curvignathus* to opportunistic pathogen using culturable aerobic bacteria isolated from termite gut and its foraging pathways.

# 2. Materials and methods

## 2.1. Preparation of culturable aerobic bacteria isolated from termite gut and its foraging pathway

Bacterial strains used in this study were collected from the gut of *C. curvignathus* and foraging pathway from three different land-use types, namely rubberwood plantation at Lembaga Getah Malaysia (RRIM) Sungai Buloh (3.16° N, 101.56° E), housing area at Puchong (2.97° N, 101.70° E) and open grassland located in Universiti Putra Malaysia, Selangor (2.98° N, 101.71° E). The detail of isolation, screening and identification was reported by Wong *et al.* [19]. The identified bacterial strains from [19] are summarized in table 1.

## 2.2 Bacterial strains and culture conditions

Bacterial strains from stock cultures were transferred to fresh Luria-Bertani (LB) (BD Difco™) agar to obtain single colony for each isolate and incubated at 28°C. Bacteria were then transferred and cultured in 500 ml Erlemeyer flasks containing 100 ml of LB broth (BD Difco™), incubated in a rotary shaker at 150 r.p.m. and maintained at a temperature of 28°C for 72 h. The suspensions of bacteria were prepared at concentrations of $10^3$, $10^6$ and $10^9$ colony-forming units (CFU) ml$^{-1}$ according to the method published by Brown [20]. At the same time, the purity and CFU for the cultures were checked to confirm its concentration. Colony morphology and Gram staining were performed as a preliminary characterization of each isolate. All procedures were performed in a biosafety cabinet with class 2 laminar flow.

## 2.3. Termites

Termites, *C. curvignathus*, were collected from termite-infested tree at the campus of Universiti Putra Malaysia, Selangor. A container with drilled holes containing rubber tree's wooden stakes was buried under the tree to bait and trap foraging termites. After 2 days, the container with infested stakes was removed and transported back to the laboratory. Debris attached on termites was separated using the bridging method as published by Tamashiro *et al.* [21].

## 2.4. Termiticidal bioassay

Two application methods were tested: (i) application of bacterial suspensions on termite via contact treatment, and (ii) application of bacterial suspensions on termite via oral-contact treatment. The contact treatment allows termites to be exposed to opportunistic pathogens through the attachment of bacteria on its body only. While oral-contact treatment allows the attachment of bacteria on the termite body and at the same time allows termite to feed in bacterial-treated filter papers. Different concentrations of bacterial suspensions: $10^3$, $10^6$ and $10^9$ CFU ml$^{-1}$ were used in the treatment. The concentrations were determined by serial dilution and plating on LB agar.

**Table 1.** Summary of bacteria identified by Wong et al. [19].

| rubberwood plantation, Sg. Buloh | | open grass area, UPM | | housing area, Puchong | |
|---|---|---|---|---|---|
| gut | foraging pathway | gut | foraging pathway | gut | foraging pathway |
| Bacillus cereus | Acintobacter baumannii | Acintobacter baumannii | Acintobacter baumannii | Acintobacter baumannii | Acintobacter baumannii |
| Acintobacter baumannii | Enterobacter cloacae | Trabulsiella guamensis | Enterobacter pulveris | Citrobacter farmeri | Pseudomonas aeruginosa |
| Citrobacter amalonaticus | Serratia marcescens | Klebsiella oxytoca | Sphingobacterium multivorum | | |
| Trabulsiella guamensis | Flavobacterium tirrenicum | Pseudomonas aeruginosa | Pseudomonas aeruginosa | | |
| Flavobacterium resinovorum | Pseudomonas aeruginosa | Microbacterium sp. | | | |
| | | Stenotrophomonas maltophilia | | | |

**Table 2.** Experimental design for the bioassay.

| no. code of bacteria | type of introduced bacteria | treatment method | bacterial concentration (CFU ml$^{-1}$) |
|---|---|---|---|
| 1 | *Acintobacter baumannii* | Feed-in (F) *or* | $10^9$ (n) *or* $10^6$ (s) *or* $10^3$ (t) |
| 2 | *Bacillus cereus* | Contact (C) | |
| 3 | *Citrobacter amalonaticus* | | |
| 4 | *Citrobacter farmeri* | | |
| 5 | *Enterobacter cloacae* | | |
| 6 | *Enterobacter pulveris* | | |
| 7 | *Flavobacterium resinovorum* | | |
| 8 | *Flavobacterium tirrenicum* | | |
| 9 | *Klebsiella oxytoca* | | |
| 10 | *Microbacterium* sp. | | |
| 11 | *Pseudomonas aeruginosa* | | |
| 12 | *Serratia marcescens* | | |
| 13 | *Sphingobacterium multivorum* | | |
| 14 | *Stenotrophomonas maltophilia* | | |
| 15 | *Trabulsiella guamensis* | | |

**Note:** Code of treatment: (treatment method)(no. code of bacteria)(bacterial concentration).
Example: code of treatment for feed-in method using *Acinetobacter baumannii* with $10^9$ concentration: F1n.

For contact treatment, 10 g of clean sand was placed in 250 ml glass jar. Each of the jars was inoculated with different concentrations ($10^3$, $10^6$ and $10^9$ CFU ml$^{-1}$) of bacterial suspensions. The sand was stirred continuously for 1 min to ensure the even spread of the bacterial suspension. Twenty-five workers and three soldier termites were placed in the jars and then covered with a thin layer of the treated sand. These termites were left in the jar containing treated sand for 30 min. These termites were then transferred to a new sterile plastic container containing moistened sterile filter papers.

For oral-contact treatment, sterile filter paper (Advantec, 55 mm diameter) was inoculated by soaking each of the filter papers with different concentrations ($10^3$, $10^6$ and $10^9$ CFU ml$^{-1}$) of bacterial suspension in a sterile glass Petri dish. This was to allow bacteria to be distributed evenly. After moistening the filter paper with bacterial suspension, treated filter papers were transferred to a sterile plastic container (55 mm diameter × 40 mm height) and allowed to air dry for 10 min. Twenty-five worker and 3 soldier termites were then placed in the plastic container containing treated filter papers.

For control samples, 25 workers and 3 soldier termites were placed in the sterile plastic container filled with moistened sterile filter paper. The filter papers act as a food source and provide moisture for termites. The experimental design is summarized in table 2. All treatments were replicated 10 times. Containers with termites were kept in an incubator at 28°C and 75% relative humidity. They were inspected daily for 7 days. Dead termites were removed for causative agent isolation work. The mortality and filter paper condition were recorded daily.

## 2.5. Reisolation of causative agent

Dead termites were removed from the container. The surfaces of fresh dead termites were sterilized with 70% ethanol. They were dissected into two parts: head with thorax and abdomen segment. The causative agent was isolated by streak plate method on LB agar and incubated at 28°C for 72 h to allow the growth of infecting bacteria. Dominant bacteria and introduced bacteria were examined in smears and Gram stain method. This was to confirm that they were identical to the bacteria which were introduced initially. These tests were in accordance with Koch's postulates as described by Bucher [22]. For live termites, they were allowed to walk on LB agar and the bodies were streaked softly on the LB agar using sterilized forceps without exposing the abdomen. Then, they were surface sterilized with 70% ethanol to remove contamination and their guts were extracted by pulling the anus with a pair of sterilized tweezers. The extracted guts were streaked on new LB agar to check the occurrence of the causative agent and introduced bacteria.

**Table 3.** Scoring of group efficacy score of mortality.

| group efficacy | Score |
|---|---|
| no effect (0) | 0 |
| slight (1–20%) | 0.2 |
| lower moderate (21–40%) | 0.4 |
| upper moderate (41–60%) | 0.6 |
| heavy (61–99%) | 0.9 |
| complete (100%) | 1 |

## 2.6. Data analysis

Mortality observed in the control samples was used to correct mortality in the treated samples according to Abbott's formula ($P = C−T/C \times 100$); $P$ = % of corrected mortality; $C$ = no. of. termites alive in the control; $T$ = no. of termites alive in the treatment [23]. The corrected mortality rates were compared among bacterial concentrations within oral-contact and contact inoculation treatment. Analysis of variance (ANOVA) SAS PROC MIXED model (SAS Institute, Cary, NC) was used to analyse the data and followed by Fisher's protected least significant difference test (LSD), to compare bacterial concentrations of an inoculation treatment. The significant levels were determined at 0.05.

The mortality percentage for each treatment (introduce bacteria × bacterial concentration) were clustered into one of the six groups: no effect (0%), slight (1–20%), lower moderate (21–40%), upper moderate (41–60%), heavy (61–99%) and complete (100%) to obtain the score of mortality which ranged from 0 to 1 (table 3). The score is assigned as such that the higher the mortality percentage, the higher the score is given. The lowest mortality score is 0 point and the highest efficacy score is 1 point; 0 point indicated non-capable to induce mortality and 1 point indicated the perfect score to induce complete mortality to *C. curvignathus*. For the score of bacteria concentrations, the scores ranged from 1 to 3. Concentrations of $10^9$ CFU ml$^{-1}$ and $10^6$ CFU ml$^{-1}$ were given 1 point and 2 points, respectively, as to rate the capability of the bacteria to induce termite mortality just by the higher concentration applied. Whereas $10^3$ CFU ml$^{-1}$ was given the highest score of 3 points as the bacteria capable to induce termite mortality even with a low concentration of bacterial suspension (table 4). Therefore, the biocontrol efficiency (%) was defined as the ability of the bacteria to induce high mortality of *C. curvignathus* with the lowest concentration of bacterial suspension at a given time under laboratory condition.

The calculation for 'biocontrol score for each bacterial concentration' and 'total biocontrol score' was calculated using equations (2.1) and (2.2), respectively. The biocontrol efficiency (%) calculation for each type of introduced bacteria was developed from equations (2.1) and (2.2) and converted into a percentage.

$$BS = MS \times CS, \tag{2.1}$$

$$TBS = BS \text{ for } 10^3 + BS \text{ for } 10^6 + BS \text{ for } 10^9 \tag{2.2}$$

and

$$BE = TBS/TG \times 100\%, \tag{2.3}$$

where:

— BS is biocontrol score for each bacterial concentration,
— TBS is total biocontrol score,
— MS is mortality,
— CS is bacterial concentration score,
— BE is biocontrol efficiency, and
— TG is total groups in the mortality score = 6 groups.

# 3. Results

## 3.1. Preliminary characterization of bacteria isolates

The isolation and characterization of bacterial colonies were observed on Petri plates after 72 h of incubation at 28°C (table 5). The growth rate for most of the isolated bacteria to become single colonies

**Table 4.** Scoring of bacteria concentration.

| bacteria concentration (CFU ml$^{-1}$) | Score |
|:---:|:---:|
| 10$^3$ | 3 |
| 10$^6$ | 2 |
| 10$^9$ | 1 |

**Table 5.** Macroscopic morphological observation of bacteria isolates.

| type of introduced bacteria | form | margin | elevation | pigment | opacity | cell shape | Gram stain |
|---|---|---|---|---|---|---|---|
| A. baumannii | round | entire | convex | creamy | opaque | rod | — |
| B. cereus | filamentous | undulate | flat | creamy | opaque | rod | + |
| C. amalonaticus | round | entire | convex | creamy | opaque | rod | — |
| C. farmer | round | entire | convex | creamy | opaque | cocci | — |
| E. cloacae | round | entire | convex | creamy | opaque | rod | — |
| E. pulveris | round | entire | convex | creamy | opaque | rod | — |
| F. resinovorum | round | entire | convex | orange | translucent | cocci | — |
| F. tirrenicum | round | entire | convex | orange | translucent | rod | — |
| K. oxytoca | round | entire | convex | grey (watery) | opaque | rod | — |
| Microbacterium sp. | round | entire | convex | creamy (waxy)[a] | opaque | cocci | + |
| P. aeruginosa | round | entire | umbonate | green | opaque | rod | — |
| S. marcescens | round | entire | convex | red | opaque | rod | — |
| S. multivorum | round | entire | convex | creamy [b] | opaque | rod | — |
| S. maltophilia | round | entire | convex | creamy | opaque | rod | — |
| T. guamensis | round | entire | convex | creamy | opaque | cocci | — |

**Note:**
[a]Bacteria turn yellow after 72 h incubation.
[b]Bacteria turn yellow after 48 h incubation.

took about 20 h, except *Microbacterium* sp., which took about 48 h to grow and turned from creamy to yellow after 72 h of the incubation period. Two isolates are shown to have crystal stain positive, which are *Bacillus cereus* and *Microbacterium* sp. Most of the colonies have round form, entire margin, convex and in rod shape. Colony elevations are convex, flat and umbonate. *Bacillius cereus* has a different colony characteristic with other bacteria which are filamentous, undulate, flat, dry texture and sometimes waxy. Colonial pigmentation of the isolates included pale creamy, orange, yellow, green, red and grey. *Flavobacterium* spp. appearance is different from the rest, with orange in colour and translucent like looking through frosted glass. *Pseudomonas aeruginosa* has a green colour with umbonate elevation. *Klebsiella oxytoca* has a watery feel when inoculate loop was used to pick up this colony. *Serratia marcescens* appears red, which becomes a significant appearance for primary identification. Sometimes, the colony has two layers of colours which are red in the middle surrounded by creamy white within a colony. *Sphingobacterium multivorum* has a similar condition with *Microbacterium* sp. which turns yellow after 48 h of incubation. Different isolates display different cell sizes and morphologies when viewed under the microscope. Some rod-shaped isolates in small or long chains were also observed.

## 3.2. Bioassay study

Pathogenicity results presented here were based on bacteria isolated from the gut of *Coptotermes curvignathus* and its foraging pathway. Termite mortality was taken at every interval of 24 h. Overall,

the results show termite mortality was low at the beginning. It gradually increased and remained constant at Day 6 and Day 7, for most of the treatments. For control samples, termites were mostly alive with low mortality percentage of 1.78%. The mortality results presented for oral-contact and contact treatments, expressed as percentages and corrected according to Abbott's formula, are shown in tables 6 and 7, respectively. The mortality was corrected using the formula so as to remove the error, if any, on account of the mortality due to factors other than the toxic effect of the types and concentrations of the introduced bacteria.

### 3.2.1. Oral-contact treatment

The oral-contact treatments, so-called because the termites had an opportunity to become infected by contacting treated filter paper as well as by eating it. The mortality results show that there was a significant difference ($p \leq 0.05$) among the concentrations on the introduced bacteria for *A. baumannii*, *B. cereus*, *E. pulveris*, *F. resinovorum*, *K. oxytoca*, *Microbacterium* sp., *P. aeruginosa*, *S. marcescens*, *S. multivorum* and *T. guamensis* (as shown in table 6). *Citrobacter amalonaticus*, *C. farmer*, *E. cloacae*, *F. tirrenicum* and *S. Maltophilia* analysis showed no significant difference ($p > 0.05$) among concentrations. This indicated that increasing the concentration from $10^3$ to $10^9$ CFU ml$^{-1}$ will not affect the rate of the mortality to any significant effect. *Serratia marcescens* and *P. aeruginosa* caused a higher mortality percentage on Day 2 regardless of the concentrations of bacterial suspension applied. With the concentration of $10^9$ CFU ml$^{-1}$, *S. marcescens*, *Microbacterium* sp., *P. aeruginosa* and *F. resinovorum* resulted in comparable mortality rate on Day 2, which showed applying $10^9$ CFU ml$^{-1}$ concentration significantly more effective than applying lower concentrations ($10^6$ and $10^3$ CFU ml$^{-1}$) for these introduced bacteria species. On Day 7, *S. marcescens* also caused a higher mortality rate (52.8%) at a low concentration of $10^3$ CFU ml$^{-1}$ when compared to other isolates. Overall, *B. cereus* caused the lowest mortality rate regardless of the concentrations applied.

Observations of termites' inactivity, inappetence or a lack of normal tactile response and termite mortality were made at 24 h intervals. With $10^9$ CFU ml$^{-1}$ concentration. *Flavobacterium resinovorum*, *S. marcescens* and *P. aeruginosa* caused higher mortality rate compared to other introduced bacteria within 48 h. This indicated that these introduced bacteria quickly cause the termite to die just by eating a small quantity of the treated filter papers, which could be due the lethal effects of the pathogen.

### 3.2.2. Contact treatment

Infection through contact method was the least effective method compared to oral-contact treatment. These data show that contact treatments for most of the bacteria acted slower and gave lower mortality, while the oral-contact method resulted in high mortality. A significant reduction ($p \leq 0.05$) in the mortality rate was observed when termites were exposed to decreased bacterial concentrations of the following introduced bacteria in the contact treatment: *B. cereus*, *E. pulveris*, *F. resinovorum*, *K. oxytoca*, *Microbacterium* sp., *P. aeruginosa*, *S. marcescens* and *S. multivorum* (table 7). While the application of different bacterial concentrations of *A. baumannii*, *C. amalonaticus*, *C. farmer*, *E. cloacae*, *F. tirrenicum*, *S. maltophilia* and *T. guamensis* in contact treatment will not cause any difference in the mortality rate of termites.

*Serratia marcescens* results showed the highest termite mortality percentage on Day 7 regardless of the bacterial concentration. *Pseudomonas aeruginosa* contact treatment results showed a similar pattern to the mortality rate data with *S. marcescens* also causing higher mortality rate within 48 h compared to other introduced bacteria. The lowest termite mortality percentages rate regardless of bacterial concentration were shown in the result data from *T. guamensis*, *K. oxytoca* and *A. baumannii*.

### 3.3. Reisolation test

Termite cadavers were dark coloured and flaccid, with the formation of a cheesy appearance and thick fluid of disintegrated tissue. The cadavers from the treatments with the bacterial concentration of $10^9$ CFU ml$^{-1}$ were sliced and separated into head and abdomen. Both parts were separately streaked on to different LB agar plates. Whereas live termites were allowed to walk on LB agar plates. The guts of those alive were extracted and streaked on LB agar plates. The naturally found bacteria in the gut of control termites were *A. baumannii*, *K. oxytoca*, *Microbacterium* sp., *P. aeruginosa*, *S. maltophilia* and *T. guamensis*.

**Table 6.** Termite mortality percentage after 7 days exposure to bacterial suspension for oral-contact treatment.

| type of introduced bacteria | bacterial concentration | average total mortality of termite by Day $n = 10$ | | | | | | | total mortality (%) | Pr > F |
|---|---|---|---|---|---|---|---|---|---|---|
| | | Day 1 | Day 2 | Day 3 | Day 4 | Day 5 | Day 6 | Day 7 | | |
| A. baumannii | $10^9$ | 2.6 | 3.9 | 3.5 | 4.2 | 4.6 | 4.9 | 5.7 | 22.8[a] | ** |
| | $10^6$ | 1.1 | 2 | 2.7 | 3.3 | 3.5 | 3.8 | 4.4 | 17.6[a] | |
| | $10^3$ | 1 | 1.3 | 1.4 | 1.4 | 1.4 | 1.9 | 2.1 | 8.4[b] | |
| B. cereus | $10^9$ | 1.1 | 1.7 | 2.8 | 3.6 | 4.1 | 4.8 | 4.9 | 19.6[a] | ** |
| | $10^6$ | 0.7 | 1.1 | 1.1 | 1.3 | 1.6 | 1.8 | 2.5 | 10[b] | |
| | $10^3$ | 0.2 | 0.2 | 0.4 | 0.7 | 1.4 | 2.2 | 2.2 | 8.8[b] | |
| C. amalonaticus | $10^9$ | 2.3 | 6.9 | 9.1 | 9.4 | 9.7 | 9.7 | 9.7 | 38.8[a] | n.s. |
| | $10^6$ | 4.6 | 6.6 | 7.2 | 7.8 | 8.3 | 8.6 | 8.6 | 34.4[a] | |
| | $10^3$ | 2.8 | 7 | 8.2 | 8.6 | 8.9 | 8.9 | 8.9 | 35.6[a] | |
| C. farmeri | $10^9$ | 2.5 | 5.9 | 7.4 | 9.3 | 11.7 | 12.4 | 12.7 | 50.8[a] | n.s. |
| | $10^6$ | 3 | 4.2 | 5.6 | 9.1 | 10.5 | 11.1 | 11.4 | 45.6[a] | |
| | $10^3$ | 3.2 | 4.7 | 6.9 | 8.2 | 9 | 9.4 | 10 | 40[a] | |
| E. cloacae | $10^9$ | 1.4 | 2.4 | 3.5 | 4.3 | 5.2 | 5.7 | 6.2 | 24.8[a] | n.s. |
| | $10^6$ | 1.4 | 2.6 | 3.4 | 3.6 | 5.2 | 5.2 | 5.7 | 22.8[a] | |
| | $10^3$ | 1.5 | 1.6 | 2.3 | 2.7 | 3.7 | 4 | 4.2 | 16.8[a] | |
| E. pulveris | $10^9$ | 2.1 | 5.7 | 8.3 | 10 | 10.4 | 10.7 | 11.1 | 44.4[a] | * |
| | $10^6$ | 1.3 | 2.3 | 6.4 | 6.7 | 8 | 8 | 8.7 | 34.8[ab] | |
| | $10^3$ | 1.2 | 1.6 | 2.8 | 3.6 | 4.4 | 4.8 | 5 | 20.0[b] | |
| F. resinovorum | $10^9$ | 2.7 | 3.4 | 4.9 | 5.1 | 6.8 | 6.8 | 7.3 | 29.2[a] | ** |
| | $10^6$ | 2.3 | 3.1 | 5.3 | 5.8 | 6.2 | 6.2 | 6.7 | 26.8[ab] | |
| | $10^3$ | 1.7 | 2.1 | 3.3 | 3.9 | 5.5 | 5.6 | 5.6 | 16.4[b] | |
| F. tirrenicum | $10^9$ | 7.2 | 8.8 | 10.9 | 11.7 | 11.8 | 12 | 12 | 48.0[a] | n.s. |
| | $10^6$ | 6.1 | 8.4 | 10.2 | 10.8 | 11.1 | 11.1 | 11.1 | 44.4[a] | |
| | $10^3$ | 3.2 | 6.2 | 8.4 | 9.2 | 10.4 | 10.6 | 10.6 | 42.4[a] | |
| K. oxytoca | $10^9$ | 2.6 | 3.4 | 3.8 | 4.6 | 4.6 | 5.2 | 5.7 | 22.8[a] | ** |
| | $10^6$ | 2.1 | 2.1 | 2.6 | 3.3 | 3.7 | 4.3 | 4.7 | 18.8[a] | |
| | $10^3$ | 1.2 | 1.2 | 1.6 | 1.6 | 1.6 | 1.7 | 2.1 | 8.4[b] | |
| Microbacterium sp. | $10^9$ | 3 | 10.9 | 11.5 | 12.3 | 13.5 | 14.5 | 16.3 | 65.2[a] | ** |
| | $10^6$ | 1.6 | 4.1 | 6.5 | 9.1 | 9.7 | 10.2 | 10.9 | 43.6[b] | |
| | $10^3$ | 1.3 | 3.0 | 5.1 | 7.4 | 9.0 | 9.8 | 10.0 | 40.0[b] | |
| P. aeruginosa | $10^9$ | 5.2 | 11.2 | 12.7 | 13.8 | 14.4 | 14.9 | 15.3 | 61.2[a] | ** |
| | $10^6$ | 5.9 | 8.5 | 10 | 10.9 | 11.7 | 11.8 | 12.1 | 48.4[b] | |
| | $10^3$ | 5.1 | 6.9 | 7.9 | 8.8 | 9.1 | 9.8 | 9.8 | 39.2[c] | |
| S. marcescens | $10^9$ | 6.1 | 11.5 | 13.8 | 15.2 | 15.8 | 17.0 | 17.0 | 68[a] | ** |
| | $10^6$ | 5.9 | 9.5 | 10.6 | 11.5 | 12.5 | 13.2 | 13.8 | 55.2[b] | |
| | $10^3$ | 6.3 | 9.0 | 9.7 | 10.0 | 11.3 | 12.3 | 13.2 | 52.8[b] | |
| S. multivorum | $10^9$ | 1.8 | 6 | 8.6 | 9 | 9.7 | 10.3 | 11.2 | 44.8[a] | ** |
| | $10^6$ | 1.5 | 6.1 | 7.8 | 8.7 | 9.3 | 9.9 | 10.2 | 40.8[a] | |
| | $10^3$ | 2 | 4.6 | 6.1 | 6.3 | 6.5 | 6.9 | 7.1 | 28.4[b] | |

(Continued.)

| type of introduced bacteria | bacterial concentration | average total mortality of termite by Day $n = 10$ | | | | | | | total mortality (%) | Pr > F |
|---|---|---|---|---|---|---|---|---|---|---|
| | | Day 1 | Day 2 | Day 3 | Day 4 | Day 5 | Day 6 | Day 7 | | |
| S. maltophilia | $10^9$ | 3.7 | 7.1 | 8.5 | 9 | 9.9 | 10.4 | 10.8 | 43.2[a] | n.s. |
| | $10^6$ | 1 | 1.6 | 5.3 | 6.9 | 8.5 | 8.5 | 8.8 | 32.5[a] | |
| | $10^3$ | 2.7 | 3.6 | 5 | 6.2 | 7.2 | 7.9 | 8.6 | 34.4[a] | |
| T. guamensis | $10^9$ | 1.5 | 2.0 | 3.4 | 3.7 | 4 | 5.5 | 6.2 | 24.8[a] | ** |
| | $10^6$ | 1.3 | 2.1 | 2.8 | 2.9 | 4.3 | 4.9 | 5.3 | 21.2[a] | |
| | $10^3$ | 0.6 | 1.2 | 1.8 | 2.4 | 2.9 | 3.1 | 3.2 | 12.8[b] | |

**Note:** Different letters between the concentrations indicate significant differences between means following LSD's test ($p \leq 0.05$).

n.s. not significant.

*Significant at $p < 0.05$.

**Significant at $p < 0.01$.

For oral-contact treatment, introduced bacteria was isolated from the heads and abdomens of cadavers. The cause of death for the termites introduced with *S. marcescens* was obviously shown on the cadavers, as the head part of the infected termite turned red, which is the signature colour of *S. marcescens* (figure 1). For surviving termites, introduced bacteria were present on the reisolation test but not as dominant bacteria.

While, for contact treatment, introduced bacteria were mostly isolated only in the abdomen of an infected cadaver. Only a few introduced bacteria (*C. farmer*, *F. tirrenicum*, *P. aeruginosa* and *S. marcescens*) were isolated on both the head and abdomen of cadavers. *Enterobacter pulveris* was not isolated in cadavers even after the application of $10^9\,\text{CFU ml}^{-1}$ bacterial concentration. Cadavers infected by *S. marcescens* via contact treatment displayed similar signs as with termites infected by *S. marcescens* via oral-contact treatment. The heads of termites turned red.

## 3.4. Biocontrol efficiency of bacteria isolated

Mortality score refers to the capability of a bacteria to kill termites in a certain bacterial concentration. This score of mortality was clustered into size groupings: no effect (0%), slight (1–20%), lower moderate (21–40%), upper moderate (41–60%), heavy (61–99%) and complete (100%) to obtain the score of mortality which ranged from 0 to 1. The score of mortality for all treatments is shown in the scatter graph (figure 2). The codes of treatments have been listed in table 2. There were only three treatments under oral-contact treatment which were ranked 'heavy' on the mortality score; *Microbacterium* sp. (code F10n), *P. aeruginosa* (code F11n) and *S. marcescens* (code F12n). All the three treatments were introduced bacteria with a concentration of $10^9\,\text{CFU ml}^{-1}$. Seven introduced bacteria with the concentration of $10^9\,\text{CFU ml}^{-1}$ in oral-contact treatment were categorized in 'upper moderate' with termite mortality percentage ranging 41–60%. None of the introduced bacteria was ranked in the group 'heavy' when the bacterial concentration reduced to $10^6$ and $10^3\,\text{CFU ml}^{-1}$ for both oral-contact and contact treatment. With the lowest bacterial concentration applied ($10^3\,\text{CFU ml}^{-1}$), there were only two introduced bacteria with oral-contact treatment that ranked under 'upper moderate' in mortality score grouping; *F. tirrenicum* and *S. marcescens*. In contact treatment, none of the treatments ranked under 'heavy' and with only four introduced bacteria (*F. tirrenicum*, *Microbacterium* sp., *P. aeruginosa* and *S. marcescens*) were ranked under 'upper moderate' with most samples inoculated with the bacterial concentration of $10^9\,\text{CFU ml}^{-1}$.

Tables 8 and 9 show the total biocontrol score and the percentage of biocontrol efficiency for each introduced bacteria via oral-contact and contact treatment. The biocontrol efficiency (%) was defined as the ability of the bacteria to induce high mortality of *C. curvignathus* with the lowest concentration of bacterial suspension at a given time under laboratory condition. In oral-contact treatment, the biocontrol efficiency of all introduced bacteria ranged from 65% to 20% with *S. marcescens* obtaining

**Table 7.** Termite mortality after 7 days exposure to bacterial suspension for contact method.

| type of introduced bacteria | bacterial concentration | mortality average of worker termites after,* | | | | | | | total mortality (%)[a] | Pr > F |
| | | Day 1 | Day 2 | Day 3 | Day 4 | Day 5 | Day 6 | Day 7 | | |
|---|---|---|---|---|---|---|---|---|---|---|
| A. baumannii | $10^9$ | 1.5 | 1.8 | 2.0 | 2.2 | 2.2 | 2.2 | 2.6 | 10.4[a] | n.s. |
| | $10^6$ | 0.4 | 0.4 | 0.6 | 0.6 | 0.6 | 1.1 | 1.4 | 5.6[a] | |
| | $10^3$ | 1 | 1.4 | 1.4 | 1.5 | 1.5 | 1.6 | 1.6 | 6.4[a] | |
| B. cereus | $10^9$ | 0.8 | 1.3 | 1.8 | 3.1 | 3.7 | 4 | 4 | 16[a] | ** |
| | $10^6$ | 0.5 | 0.5 | 0.5 | 1 | 1 | 1.1 | 1.6 | 6.4[b] | |
| | $10^3$ | 0.5 | 1.1 | 1.1 | 1.4 | 1.6 | 1.6 | 1.7 | 6.8[b] | |
| C. amalonaticus | $10^9$ | 2 | 5 | 7.2 | 8.4 | 8.4 | 9 | 9 | 36.0[a] | n.s. |
| | $10^6$ | 2.4 | 5 | 6.8 | 7.2 | 7.6 | 7.8 | 7.8 | 31.2[a] | |
| | $10^3$ | 2.6 | 6 | 7.4 | 7.8 | 7.8 | 8 | 8.2 | 32.8[a] | |
| C. farmeri | $10^9$ | 2.1 | 3.2 | 4.1 | 6.2 | 7.5 | 9.5 | 9.7 | 38.8[a] | n.s. |
| | $10^6$ | 3.9 | 5.2 | 5.7 | 6.1 | 6.8 | 7.4 | 8.5 | 34[a] | |
| | $10^3$ | 4.3 | 6.9 | 7.4 | 7.7 | 8.1 | 8.4 | 8.9 | 35.6[a] | |
| E. cloacae | $10^9$ | 1.1 | 2.2 | 2.4 | 2.6 | 2.6 | 3.1 | 3.3 | 13.2[a] | n.s. |
| | $10^6$ | 0.8 | 1.3 | 1.8 | 2.2 | 2.4 | 2.8 | 3.4 | 13.6[a] | |
| | $10^3$ | 1.1 | 1.3 | 1.5 | 1.7 | 2 | 2 | 2.2 | 8.8[a] | |
| E. pulveris | $10^9$ | 2.9 | 5.2 | 6.8 | 7.2 | 7.4 | 7.7 | 7.9 | 31.6[a] | ** |
| | $10^6$ | 2.6 | 5.6 | 6.1 | 6.4 | 6.9 | 7.3 | 7.3 | 29.2[a] | |
| | $10^3$ | 2.9 | 4.4 | 4.6 | 4.7 | 4.8 | 4.8 | 4.8 | 19.2[b] | |
| F. resinovorum | $10^9$ | 0.8 | 2.0 | 2.2 | 2.5 | 2.8 | 2.8 | 3.4 | 13.5[a] | ** |
| | $10^6$ | 0.8 | 1.3 | 2.2 | 2.4 | 2.4 | 2.4 | 2.4 | 9.6[b] | |
| | $10^3$ | 0.6 | 0.6 | 1.0 | 1.0 | 1.2 | 1.2 | 1.2 | 4.8[b] | |
| F. tirrenicum | $10^9$ | 3.9 | 8.1 | 10 | 10 | 10.3 | 10.3 | 10.3 | 41.2[a] | n.s. |
| | $10^6$ | 3.8 | 7.6 | 8.8 | 9.2 | 9.6 | 9.6 | 9.6 | 38.4[a] | |
| | $10^3$ | 3.6 | 6.7 | 8.2 | 8.5 | 8.5 | 8.5 | 8.5 | 34.0[a] | |
| K. oxytoca | $10^9$ | 0.8 | 1.8 | 2.1 | 2.3 | 2.5 | 2.7 | 2.7 | 10.8[a] | ** |
| | $10^6$ | 0.1 | 0.6 | 0.8 | 1.0 | 1.0 | 1.1 | 1.1 | 4.4[b] | |
| | $10^3$ | 0.7 | 0.8 | 0.9 | 0.9 | 1.0 | 1.1 | 1.1 | 4.4[b] | |
| Microbacterium sp. | $10^9$ | 2.9 | 7.1 | 9.9 | 10 | 10.4 | 11 | 11.1 | 44.4[a] | ** |
| | $10^6$ | 1.1 | 2.1 | 2.2 | 2.4 | 5.7 | 6.2 | 6.4 | 25.6[b] | |
| | $10^3$ | 0.8 | 1.1 | 1.3 | 1.4 | 2.4 | 3.0 | 2.9 | 11.6[c] | |
| P. aeruginosa | $10^9$ | 4.8 | 7.2 | 10.8 | 11.2 | 11.8 | 12.1 | 12.4 | 49.6[a] | ** |
| | $10^6$ | 3.7 | 6.9 | 10.4 | 10.4 | 10.7 | 10.7 | 10.7 | 42.8[a] | |
| | $10^3$ | 3.1 | 5.8 | 7.8 | 7.8 | 7.8 | 8.1 | 8.1 | 32.4[b] | |
| S. marcescens | $10^9$ | 5.4 | 11.1 | 11.3 | 12.2 | 12.8 | 13.1 | 13.4 | 53.6[a] | ** |
| | $10^6$ | 4.8 | 10.8 | 11.2 | 11.4 | 11.8 | 12.4 | 12.8 | 51.2[a] | |
| | $10^3$ | 2.6 | 4.4 | 7.4 | 7.6 | 8.0 | 8.0 | 8.0 | 32.0[b] | |
| S. multivorum | $10^9$ | 3.9 | 5.7 | 6.1 | 7.2 | 7.8 | 7.8 | 8.0 | 32.0[a] | ** |
| | $10^6$ | 1 | 2 | 3.5 | 4.8 | 5.1 | 5.5 | 6.2 | 24.8[ab] | |
| | $10^3$ | 1.2 | 1.8 | 1.8 | 2.7 | 3.2 | 3.4 | 4.7 | 18.8[b] | |

(Continued.)

**Table 7.** (Continued.)

| type of introduced bacteria | bacterial concentration | mortality average of worker termites after,* | | | | | | | total mortality (%)[a] | Pr > F |
|---|---|---|---|---|---|---|---|---|---|---|
| | | Day 1 | Day 2 | Day 3 | Day 4 | Day 5 | Day 6 | Day 7 | | |
| S. maltophilia | $10^9$ | 1.8 | 2.6 | 2.7 | 2.9 | 3.2 | 3.4 | 3.5 | 14[a] | n.s. |
| | $10^6$ | 1.3 | 2.4 | 2.7 | 3.1 | 3.4 | 3.4 | 3.4 | 13.6[a] | |
| | $10^3$ | 0.6 | 1.8 | 2.2 | 2.6 | 2.7 | 2.9 | 2.9 | 11.6[a] | |
| T. guamensis | $10^9$ | 1.5 | 1.6 | 1.9 | 2.3 | 2.6 | 2.6 | 2.7 | 10.8[a] | n.s. |
| | $10^6$ | 0 | 0.3 | 0.7 | 1 | 1.2 | 1.5 | 1.5 | 6[a] | |
| | $10^3$ | 0.2 | 0.5 | 0.8 | 1.3 | 1.4 | 1.7 | 1.7 | 6.8[a] | |

**Note:** Different letters between the concentrations indicate significant differences between means following LSD's test ($p \leq 0.05$). n.s. not significant.
*Corrected by Abbott's formula.
**Significant at $p < 0.01$.

highest score of 65% (table 8). This is followed by *F. tirrenicum* (60%), *Microbacterium* sp. (55%) and *P. aeruginosa* (55%). The lowest biocontrol score was obtained by *B. cereus*.

The biocontrol efficiency for contact treatment ranged from 20% to 50%. *Serratia marcescens* and *P. aeruginosa* obtained the highest biocontrol score of 50% via contact treatment. This was followed by *Microbacterium* sp. and *F. tirrenicum* with biocontrol efficiency of 43.3%. *Flavobacterium resinovorum*, *S. maltophilia A. baumannii*, *B. cereus*, *E. cloacae*, *K. oxytoca* and *T. guamensis* in contact method obtained the lowest biocontrol efficiency of 20%.

# 4. Discussion

The present study shows the ability of bacteria associated in the gut of *C. curvignathus* and its foraging area as an opportunistic pathogen against this pest in laboratory scale. It is not easy to infect termite with the low concentration of bacteria or pathogen because of termites' natural behaviour of mutual grooming to remove pathogen among nest-mates. Mutualistic association between termites and their dense and diverse bacterial gut microbiota gives them a strong immune system via opsonization, phagocytosis, melanization, nodulation, encapsulation and coagulation to kill pathogen once they enter termite digestive gut [24].

The results show an increase in the mortality rate of termites when exposed to bacteria by oral-contact and contact treatments. In most cases, entomopathogenic bacteria attack their host after ingestion and cause disruption of gut tissue, followed by septicaemia [9]. From this study, it was observed that infection by contact was the least effective method compared to oral-contact treatment. These data show that by contact treatments, most of the bacteria acted slower and gave lower mortality, while the oral-contact treatment resulted in higher mortality. The low rate of mortality in the contact treatment for most bacteria may be due to the inability of the bacterium to penetrate the integument of *C. curvignathus*.

Differences were recorded through oral-contact and contact treatment methods. We assumed contact treatment will have no significant impact on termite mortality due to the grooming behaviour of termites, and unlike fungi, bacteria do not have hyphae to readily penetrate the insect exoskeleton. However, results of contact treatment from some of the bacteria such as *S. marcescens* and *P. aeruginosa* gave a significant impact on termite's mortality rate. It assumed to start with the attachment of the bacteria to the surface of termites through contact treatment. The mutual grooming behaviour and eating the foreign organisms as an act to remove it from the body surface of its nest-mates has hastened the spread of disease through a population. The function of grooming in termites has always been to replenish their supply of bacteria in the gut, and to remove foreign bodies such as bacteria or parasites from the integument and eliminate through alimentary tracts. However, abnormally high bacterial concentration introduced may cause disruption to their digestive system, by disrupting mutual interactions between termite hosts and their symbionts [25]. Previous reports showed that grooming increases with the presence of several

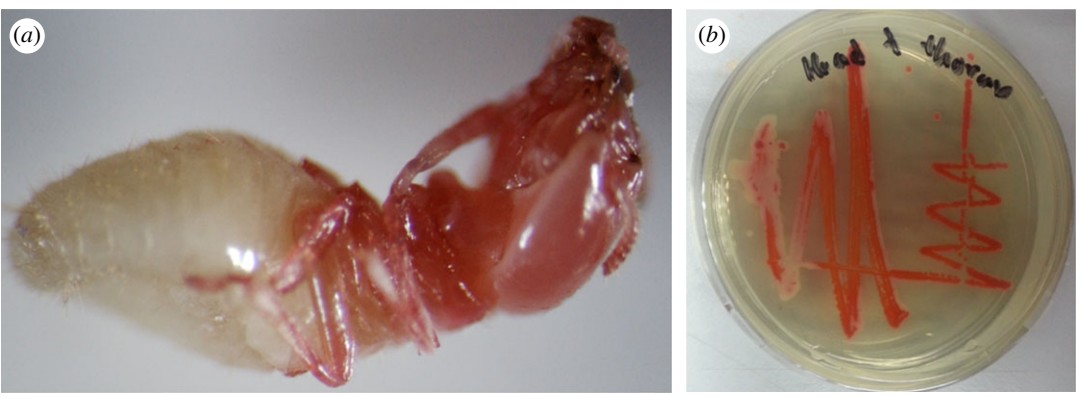

**Figure 1.** (*a*) Termite infected with *Serratia marcescens* after 3 days; (*b*) *Serratia marcescens* reisolated from head and thorax part of infected termite.

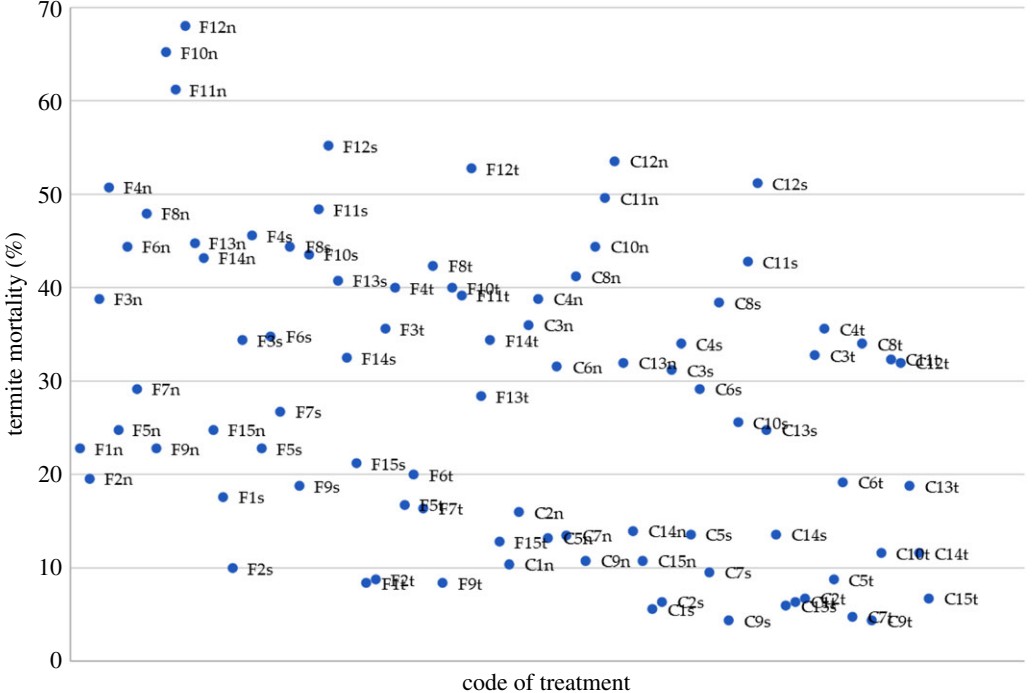

**Figure 2.** Bacteria efficacy grouping of inoculation treatments (treatment code according to table 2).

entomopathogens on the integument of termites [26]. So even through contact treatment, the termites would have ingested the introduced bacteria through mutual grooming, but probably a lower bacterial concentration was ingested when compared with the same bacterial concentration inoculated through oral-contact treatment. This is shown in the higher mortality rate in oral-contact treatment compared to contact treatment when the same bacterial concentration was introduced to the termites.

Termites were highly susceptible to *S. marcescens*, both contact and oral-contact treatment. *Serratia marcescens* is a Gram-negative bacterium that is unable to produce spores. *Serratia marcescens* has been reported to infect termites and cause septicaemia [27–29]. The ability of *S. marcescens* to produce red pigment has become a marker to trace bacterial activity. Due to the presence of *S. marcescens*, a large number of the dead termites exhibited a red discoloration and apparently was the cause of the high initial mortality among samples introduced with the bacterium: both oral-contact and contact treatments. Similar symptoms of infected bacteria were also reported by Wang *et al*. [29]. As shown in this study, *S. marcescens* with its entomopathogenic ability is possibly due to the present of pADAP protein in the bacteria [30]. As response to the release of this bacterial protein, termites expulse the gut contents to the hindgut and release frass pellets [31]. Eventually, this causes the weakening of internal tissues of termite, which may lead to bacterial incursion of the haemocoel, causing septicaemia and death [31].

**Table 8.** Biocontrol score for oral-contact treatment. Types of introduced bacteria were arranged from the most effective to the weakest.

| type of introduced bacteria | biocontrol score of CFU ml$^{-1}$ | | | total biocontrol score | biocontrol efficiency (%) |
| --- | --- | --- | --- | --- | --- |
| | $10^3$ | $10^6$ | $10^9$ | | |
| S. marcescens | 0.6 | 0.6 | 0.9 | 3.9 | 65.0 |
| F. tirrenicum | 0.6 | 0.6 | 0.6 | 3.6 | 60.0 |
| Microbacterium sp. | 0.4 | 0.6 | 0.9 | 3.3 | 55.0 |
| P. aeruginosa | 0.4 | 0.6 | 0.9 | 3.3 | 55.0 |
| C. farmeri | 0.4 | 0.6 | 0.6 | 3.0 | 50.0 |
| S. multivorum | 0.4 | 0.6 | 0.6 | 3.0 | 50.0 |
| E. pulveris | 0.2 | 0.4 | 0.6 | 2.0 | 43.3 |
| S. maltophilia | 0.4 | 0.4 | 0.6 | 2.6 | 43.3 |
| C. amalonaticus | 0.4 | 0.4 | 0.4 | 2.4 | 40.0 |
| F. resinovorum | 0.2 | 0.4 | 0.4 | 1.8 | 30.0 |
| T. guamensis | 0.2 | 0.4 | 0.4 | 1.8 | 30.0 |
| E. cloacae | 0.2 | 0.4 | 0.4 | 1.8 | 30.0 |
| A. baumannii | 0.2 | 0.2 | 0.4 | 1.4 | 23.3 |
| K. oxytoca | 0.2 | 0.2 | 0.4 | 1.4 | 23.3 |
| B. cereus | 0.2 | 0.2 | 0.2 | 1.2 | 20.0 |

**Table 9.** Biocontrol score for contact method. Types of introduced bacteria were arranged from the most effective to the weakest.

| type of introduced bacteria | biocontrol score of CFU ml$^{-1}$ | | | biocontrol score | efficiency biocontrol (%) |
| --- | --- | --- | --- | --- | --- |
| | $10^3$ | $10^6$ | $10^9$ | | |
| S. marcescens | 0.4 | 0.6 | 0.6 | 3.0 | 50.0 |
| P. aeruginosa | 0.4 | 0.6 | 0.6 | 3.0 | 50.0 |
| Microbacterium sp. | 0.2 | 0.4 | 0.6 | 2.0 | 43.3 |
| F. tirrenicum | 0.4 | 0.4 | 0.6 | 2.6 | 43.3 |
| C. amalonaticus | 0.4 | 0.4 | 0.4 | 2.4 | 40.0 |
| C. farmeri | 0.4 | 0.4 | 0.4 | 2.4 | 40.0 |
| S. multivorum | 0.2 | 0.4 | 0.4 | 1.8 | 30.0 |
| E. pulveris | 0.2 | 0.4 | 0.4 | 1.8 | 30.0 |
| F. resinovorum | 0.2 | 0.2 | 0.2 | 1.2 | 20.0 |
| S. maltophilia | 0.2 | 0.2 | 0.2 | 1.2 | 20.0 |
| A. baumannii | 0.2 | 0.2 | 0.2 | 1.2 | 20.0 |
| B. cereus | 0.2 | 0.2 | 0.2 | 1.2 | 20.0 |
| E. cloacae | 0.2 | 0.2 | 0.2 | 1.2 | 20.0 |
| K. oxytoca | 0.2 | 0.2 | 0.2 | 1.2 | 20.0 |
| T. guamensis | 0.2 | 0.2 | 0.2 | 1.2 | 20.0 |

Besides, *S. marcescens* has a strong chitinase mechanism and produces high levels of chitinolytic enzymes which are capable to breakdown the chitin in the insect exoskeleton. The study by Bahar *et al.* [32] showed a very strong positive connection between the chitinase activities and the insecticidal

activities of *S. marcescens*. The expressed chitinases ChiA, ChiB and ChiC1 from *S. marcescens* showed toxicity against the honeybee mite, *V. destructor* in the laboratory [33]. Bahar *et al.* [32] found *S. marcescens* effective in killing coleopteran insects with more chitin in their exoskeleton. It was reported by Lundgren & Jurat-Fuentes [34] that termite has a weak point which may allow the bacterial entry related with *S. marcescens* infection at the foregut–midgut junction, although further study would be required to confirm this. Foregut and hindgut of termite are lined with chitinous cuticle, but it is absent in the midgut, which is lined by peritrophic membrane, a matrix of secreted protein fibrils, proteoglycans and glycoproteins function as physical and immunological protection [34]. *Serratia marcescens* lethality to insect hosts has been shown to have proteolytic and chitinolytic virulence factors that degrade the peritrophic membrane [35]. The zone of transition from cuticular lining to peritrophic membrane may present a vulnerability to infection which can be exploited by certain bacteria. This could explain the high mortality rate of termites introduced with *S. marcescens* even just by introducing the bacterium through contact treatment.

*Pseudomonas aeruginosa* is a Gram-negative bacterium that can inhibit the growth of a wide range of organisms and pathogenic to mammals [36], insects [37], nematodes [38] and plants [39]. Data showed termites to be highly susceptible to *P. aeruginosa* at $10^9$ CFU ml$^{-1}$ via oral-contact treatment. Similar results have been reported by Khan *et al.* [40], who tested the pathogenicity of *P. aeruginosa* against *Mordellistena championi*, *Heterotermes indicola* and *Coptotermes heimi* (Wasmann) (Rhinotermitidae) in the laboratory, where termite mortality ranged from 25 to 52%. Another similar report by Wang *et al.* [41] found termite mortality ranged from 40 to 50% for 12 days against *Coptotermes formosanus*. Hydrogen cyanide is a secondary metabolite of P. aeruginosa [42], which is believed to be responsible for the entomopathogenic potential of this bacteria [43,44]. Devi & Kothamasi [45] demonstrated that cyanide of bacterial origin may constrain cytochrome c oxidase (CCO) of the termite *Odontotermes obesus* respiratory chain and proved that *Pseudomonas* spp., hydrogen cyanide-producing bacteria, can actually kill a macroscopic insect pest by cyanide poisoning. This ability of pseudomonad metabolites such as cyanide represents an attractive possibility for insect pest management as it may overcome the behavioural adaptations of social insect pests such as termites, by blocking the termite respiration instead of through infection or predation. This may explain the considerably high mortality rate of termites introduced with *P. aeruginosa* even via contact treatment. The reisolation test results on the cadaver and termites introduced with *P. aeruginosa* via oral-contact and contact treatment, also shows that *P. aeruginosa* was not the dominating bacteria and the signature colour of the bacteria; yellow or green was not seen on the cadaver or termites treated with the bacteria.

Bacteria isolated from termite gut such as *A. baumanni*, *B. cereus*, *F. resinovorum*, *K. oxytoca* and *T. guamensis* were less pathogenic to termite, recording a mortality percentage of approximately 20%. The long-standing association between termites and prokaryotic microorganisms plays an important role especially in food digesting. These bacteria are seldom harmful or pathogenic towards the host and hence not potentially useful as biological control agent bacteria [46]. Normally, gut microbiota plays a role in detoxification of food. The presences of these bacteria are not only for the degradation of food but also neutralization of toxin in food [47]. Therefore, termites would not try to get rid of such beneficial bacteria unless the bacteria have reached a certain level of concentration that could cause death. For example, *Klebsiella oxytoca* under $10^6$ CFU ml$^{-1}$ caused 32.8% mortality percentage through oral-contact treatment, whereas $10^3$ CFU ml$^{-1}$ gave 18.4% mortality percentage. *Klebsiella oxytoca* can be claimed as beneficial bacteria because of their ability to fix nitrogen [48] and break down cellulose [49]. Nevertheless, bacteria concentrations do not play an important role in causing death through contact method. This was implied by most bacteria showing no significant difference among the three concentrations.

*Serratia marcescens* was found dominated at the head and thorax parts of termites introduced with the bacteria. However, the bacterium did not dominate the termites' gut. The gut microbiota in termites is believed to inhibit the bacterial replication, thus precluding *S. marcescens* invasion in the gut. The viability of bacteria may decline after being in the alimentary tracts of the termites. Veivers *et al.* [50] showed an increase in the numbers of the normal gut microbiota in the gut of *Coptotermes lacteus* when exposed to foreign bacteria. They also stated that the increase of normal gut microbiota is important in controlling foreign bacteria from invading. Competition with normal termite gut flora appears to be the main reason why experimentally introduced bacteria are not retained or dominating during the experiment. Work by Husseneder *et al.* [51] confirms that bacteria such as *Escherichia coli*, when introduced to termites with normal gut flora, are unable to survive and lose the ability to be cultured over time. Nonetheless, bacteria were efficiently introduced/inoculated, as many of the foreign bacteria that do not exist in natural termite gut were present. Although the experimental design did not measure the modes of transfer among termites, most likely bacteria were transferred by grooming, trophallaxis and/or coprophagy [52].

From the mortality percentage, we ranked each bacterium according to the obtained biocontrol score and its calculated efficiency biocontrol (%). From the efficiency biocontrol, the most efficient bacteria to control termite survival through oral-contact treatment are *S. marcescens* (65%). This is followed by, *F. tirrenicum* (60%), *Microbacterium* sp. (55%) and *P. aeruginosa* (55%). As for contact treatment, *S. marcescens* (50%) and *P. aeruginosa* (50%) are the most effective isolates in controlling termite among the 15 bacteria. Bacteria are able to attach on termite body and probably ingested through mutual grooming. The consistency in termite mortality for all concentrations suggested that *F. tirrenicum* was able to disrupt termite activity and cause death. To date, there has been no written documentation on using *F. tirrenicum* and *Microbacterium* sp. as a termite biocontrol agent. *Microbacterium* spp. was only known to cause intestinal inflammation in nematodes. The actinobacterium *Microbacterium* sp. has been reported to be associated with the gut of termite [53,54] and play a role in lignin degradation [52]. In this study, termites were also highly susceptible to *Microbacterium* sp. via oral-contact treatment. *Microbacterium* sp. has the ability to survive for a long time within the host in non-replicating and drug-tolerant state and later awaken to cause disease [55]. However, the cause of termite death due to the inoculation of *Microbacterium* sp. and *F. tirrenicum* is unknown and further studies are required.

# 5. Conclusion

The present study determined the susceptibility of *Coptotermes curvignathus* towards opportunistic pathogen using culturable aerobic bacteria isolated from termite gut and its foraging pathways. Oral-contact treatment resulted in a higher mortality rate of termites. In conclusion, we can point out a few potential statements through this study, (i) *Microbacterium* sp. and *F. tirrenicum*, which has not been discovered of its potential as pathogen agent, can be further studied on its susceptibility to various host, (ii) bacteria can kill termite even through physical contact with infected area, (iii) septicaemia in termites is still inevitable when a high level of bacteria concentration is introduced through feed, and (iv) microbes are able to disturb termite gut ecology under favourable condition. The results of this study indicate that *P. aeruginosa* and *S. marcescens* in particular may be attractive candidates worth further examination as a possible biocontrol agent against *Coptotermes curvignathus*. However, further research is required to determine the efficiency of these bacteria in the field. Moreover, mass-produced formulation studies and environmental and ecological risk evaluation of the potential biocontrol are also needed.

Ethics. We declare that the work submitted for the publication is original and has not been published elsewhere, accepted for publication elsewhere or under editorial review for publication elsewhere; and that all the authors mutually agree with its content and have approved the paper for release and submission. All the authors have declared no conflict of interest. The manuscript does not contain experiments using animals. At the same time, the manuscript does not contain human studies.

Data accessibility. Data available from the Dryad Digital Repository at: https://doi.org/10.5061/dryad.2ngf1vhk8 [56].

Authors' contributions. K.L.C. carried out data collection, participated in analysis and interpretation of data and drafted the article; P.S.H. agreed to be accountable for all aspects of the work in ensuring that questions related to the accuracy or integrity of any part of the work are appropriately investigated and resolved; A.C.L. and Z.A. edited and revised the article critically for important intellectual content; C.L.L. participated in analysis and interpretation of data; P.S.K. participated in analysis and interpretation of data; W.Z.W. and S.G. carried out data collection. All authors gave final approval for publication.

Competing interests. We have no competing interests.

Funding. The authors are grateful for the financial support from co-author P.S.H. under the Higher Institution Centre of Excellence (HICoE) (grant no. 6369110) project at the Institute of Tropical Forestry and Forest Products which was given by the Ministry of Higher Education Malaysia (MOHE). This research was also supported by the Fundamental Research Grant Scheme (FRGS) from the Ministry of Higher Education (MOHE), Malaysia under the grant FRGS/1/2019/WAB07/UPM/02/2.

Acknowledgements. The authors also thank all project members for support and collaboration. The authors also sincerely thank the postgraduate students that participated in the field sampling exercise.

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
