## [Reviewer comments · Royal Society Open Science]

Review History

RSOS-200847.R0 (Original submission)

Review form: Reviewer 1

Is the manuscript scientifically sound in its present form?

Yes

Are the interpretations and conclusions justified by the results?

Yes

Is the language acceptable?

Yes

Do you have any ethical concerns with this paper?

No

Have you any concerns about statistical analyses in this paper?

No

Recommendation?

Accept with minor revision (please list in comments)

Comments to the Author(s)

The manuscript presents a research analysis showing that *P. aeruginosa* and *S. marcescens* in particular may be attractive candidates worth further testing as a possible biological control agent against *C. curvignathus* in the field and to assess the environmental and ecological hazards associated with biological control.

General the manuscript contains new information and analysis, also conclusions briefly but adequately supported by the presented data.

In my opinion the manuscript has high scientific value. Authors presented an interesting introduction to the topic, described the research methodology in detail, the results discussion is extensive (very well written). The overall structure and idea in the revised paper are very interesting. In my opinion this manuscript can be published in Royal Society Open Science, minor corrections need to be made - suggestions below:

- at the summary it is good to define the scientific question - Authors can optional consider the possibility of implementing this issue;
- Authors also may consider changing the form of writing equations 1, 2 and 3 (page 9, line 20-23) - it will be much more visible if replace the descriptions provided in equations by entering symbols and adding symbol descriptions (as in the form of presenting typical equations);
- in the manuscript, Authors did not include Table 8, after Table 7, the next one is Table 9. Finally, there is an attached Table 8, to which Authors do not refer in the text.

Review form: Reviewer 2

Is the manuscript scientifically sound in its present form?

Yes

Are the interpretations and conclusions justified by the results?

Yes

Is the language acceptable?

Yes

Do you have any ethical concerns with this paper?

No

Have you any concerns about statistical analyses in this paper?

No

Recommendation?

Accept with minor revision (please list in comments)

Comments to the Author(s)

The manuscript contains valuable results that can be useful in biocontrol of termites population. The manuscript is written neatly and easy to understand. Well design experiments were performed. Methods are adequate and appropriate statistical analysis make the results reliable. It's a good and ripe paper.

The one change I would suggest is re-edition of Table 2.

Table 2 is too abundant and takes to much space in a paper. I would consider to re-edit/simplify it AND present the data from table 2 in full as a supplementary material.

When that minor revision is done, paper is ready for publication.

Decision letter (RSOS-200847.R0)

Dear Dr LEE

On behalf of the Editors, I am pleased to inform you that your Manuscript RSOS-200847 entitled "Septicemia of Subterranean Termites *Coptotermes curvignathus* Caused by Disturbance of Bacteria Isolated from Termite Gut and Its Foraging Pathways" has been accepted for publication in Royal Society Open Science subject to minor revision in accordance with the referee suggestions. Please find the referees' comments at the end of this email.

The reviewers and handling editors have recommended publication, but also suggest some minor revisions to your manuscript. Therefore, I invite you to respond to the comments and revise your manuscript.

- Ethics statement

- Data accessibility

<http://datadryad.org/submit?journalID=RSOS&manu=RSOS-200847>

- Competing interests

- Authors' contributions

- Acknowledgements

- Funding statement

Because the schedule for publication is very tight, it is a condition of publication that you submit the revised version of your manuscript before 26-Jul-2020. Please note that the revision deadline will expire at 00.00am on this date. If you do not think you will be able to meet this date please let me know immediately.

- 1) A text file of the manuscript (tex, txt, rtf, docx or doc), references, tables (including captions) and figure captions. Do not upload a PDF as your "Main Document";
- 2) A separate electronic file of each figure (EPS or print-quality PDF preferred (either format should be produced directly from original creation package), or original software format);
- 3) Included a 100 word media summary of your paper when requested at submission. Please ensure you have entered correct contact details (email, institution and telephone) in your user account;
- 4) Included the raw data to support the claims made in your paper. You can either include your data as electronic supplementary material or upload to a repository and include the relevant doi

within your manuscript. Make sure it is clear in your data accessibility statement how the data can be accessed;

5) All supplementary materials accompanying an accepted article will be treated as in their final form. Note that the Royal Society will neither edit nor typeset supplementary material and it will be hosted as provided. Please ensure that the supplementary material includes the paper details where possible (authors, article title, journal name).

If your manuscript is newly submitted and subsequently accepted for publication, you will be asked to pay the article processing charge, unless you request a waiver and this is approved by Royal Society Publishing. You can find out more about the charges at <https://royalsocietypublishing.org/rsos/charges>. Should you have any queries, please contact openscience@royalsociety.org.

on behalf of Dr John Dalton (Associate Editor) and Pete Smith (Subject Editor)
openscience@royalsociety.org

Associate Editor Comments to Author (Dr John Dalton):

The manuscript was detailed, informative and novel. The reviewers suggest several alterations to Tables and Figures to improve the presentation. Please check order of Tables and consider alterations suggested by reviewers that may make some clearer to the reader.

Reviewer comments to Author:
Reviewer: 1

Comments to the Author(s)

The manuscript presents a research analysis showing that *P. aeruginosa* and *S. marcescens* in particular may be attractive candidates worth further testing as a possible biological control agent against *C. curvignathus* in the field and to assess the environmental and ecological hazards associated with biological control.

General the manuscript contains new information and analysis, also conclusions briefly but adequately supported by the presented data.

In my opinion the manuscript has high scientific value. Authors presented an interesting introduction to the topic, described the research methodology in detail, the results discussion is

extensive (very well written). The overall structure and idea in the revised paper are very interesting. In my opinion this manuscript can be published in Royal Society Open Science, minor corrections need to be made - suggestions below:

- at the summary it is good to define the scientific question - Authors can optional consider the possibility of implementing this issue;
- Authors also may consider changing the form of writing equations 1, 2 and 3 (page 9, line 20-23) - it will be much more visible if replace the descriptions provided in equations by entering symbols and adding symbol descriptions (as in the form of presenting typical equations);
- in the manuscript, Authors did not include Table 8, after Table 7, the next one is Table 9. Finally, there is an attached Table 8, to which Authors do not refer in the text.

Reviewer: 2

Comments to the Author(s)

The manuscript contains valuable results that can be useful in biocontrol of termites population. The manuscript is written neatly and easy to understand. Well design experiments were performed. Methods are adequate and appropriate statistical analysis make the results reliable. It's a good and ripe paper.

The one change I would suggest is re-edition of Table 2.

Table 2 is too abundant and takes to much space in a paper. I would consider to re-edit/simplify it AND present the data from table 2 in full as a supplementary material.

When that minor revision.is done, paper is ready for publication.

Author's Response to Decision Letter for (RSOS-200847.R0)

See Appendix A.

Decision letter (RSOS-200847.R1)

Dear Dr Lee,

It is a pleasure to accept your manuscript entitled "Septicemia of Subterranean Termites *Coptotermes curvignathus* Caused by Disturbance of Bacteria Isolated from Termite Gut and Its Foraging Pathways" in its current form for publication in Royal Society Open Science.

Best regards,

on behalf of Dr John Dalton (Associate Editor) and Pete Smith (Subject Editor)
openscience@royalsociety.org

Appendix A

RESPONSE TO REVIEWER COMMENTS

Septicemia of Subterranean Termites *Coptotermes curvignathus* Caused by Disturbance of Bacteria Isolated from Termite Gut and Its Foraging Pathways

Thank you for the thoughtful reviews of our manuscript. We take concerns seriously and have addressed them to the best of our abilities. Changes have been made as suggested by the reviewers and were highlighted in yellow in the manuscript. Some of the more notable changes are listed as below;

1) at the summary it is good to define the scientific question - Authors can optionally consider the possibility of implementing this issue. (Reviewer #1)

✓ Changes have been made as suggested by the reviewers and were highlighted in yellow in the manuscript.

2) Authors also may consider changing the form of writing equations 1, 2 and 3 (page 9, line 20-23) - it will be much more visible if replace the descriptions provided in equations by entering symbols and adding symbol descriptions (as in the form of presenting typical equations). (Reviewer #1)

✓ Changes have been made as suggested by the reviewers and were highlighted in yellow in the manuscript.

3) In the manuscript, Authors did not include Table 8, after Table 7, the next one is Table 9. Finally, there is an attached Table 8, to which Authors do not refer in the text. (Reviewer #1)

✓ Correction has been made. The numbering of the tables has been corrected.

4) Table 2 is too abundant and takes too much space in a paper. I would consider to re-edit/simplify it AND present the data from table 2 in full as a supplementary material. (Reviewer #2)

✓ Correction has been made as suggested by the reviewer. Table 2 was simplified to make it easier to understand and to enhance readability.